# Understanding the influence of socioeconomic status on the association between combinations of lifestyle factors and adverse health outcomes: a systematic review protocol

Hamish Foster ,[1] Peter Polz ,[1] Frances Mair,[1] Jason Gill,[2]
Catherine A O'Donnell [1]

¹General Practice and Primary Care, Institute of Health and Wellbeing, University of Glasgow, Glasgow, UK
²Institute of Cardiovascular and Medical Sciences, University of Glasgow, Glasgow, UK

**Correspondence to**
Professor Catherine A O'Donnell;
Kate.O'Donnell@glasgow.ac.uk

## ABSTRACT

**Introduction** Combinations of unhealthy lifestyle factors are strongly associated with mortality, cardiovascular disease (CVD) and cancer. It is unclear how socioeconomic status (SES) affects those associations. Lower SES groups may be disproportionately vulnerable to the effects of unhealthy lifestyle factors compared with higher SES groups via interactions with other factors associated with low SES (eg, stress) or via accelerated biological ageing. This systematic review aims to synthesise studies that examine how SES moderates the association between lifestyle factor combinations and adverse health outcomes. Greater understanding of how lifestyle risk varies across socioeconomic spectra could reduce adverse health by (1) identifying novel high-risk groups or targets for future interventions and (2) informing research, policy and interventions that aim to support healthy lifestyles in socioeconomically deprived communities.

**Methods and analysis** Three databases will be searched (PubMed, EMBASE, CINAHL) from inception to March 2020. Reference lists, citations and grey literature will also be searched. Inclusion criteria are: (1) prospective cohort studies; (2) investigations of two key exposures: (a) lifestyle factor combinations of at least three lifestyle factors (eg, smoking, physical activity and diet) and (b) SES (eg, income, education or poverty index); (3) an assessment of the impact of SES on the association between combinations of unhealthy lifestyle factors and health outcomes; (4) at least one outcome from—mortality (all cause, CVD and cancer), CVD or cancer incidence. Two independent reviewers will screen titles, abstracts and full texts of included studies. Data extraction will focus on cohort characteristics, exposures, direction and magnitude of SES effects, methods and quality (via Newcastle-Ottawa Scale). If appropriate, a meta-analysis, pooling the effects of SES, will be performed. Alternatively, a synthesis without meta-analysis will be conducted.

**Ethics and dissemination** Ethical approval is not required. Results will be disseminated via peer-reviewed publication, professional networks, social media and conference presentations.

**PROSPERO registration number** CRD42020172588.

## Strengths and limitations of this study

► This review protocol lays out a comprehensive search strategy and a rigorous data extraction and synthesis plan to collate the evidence for the effect of socioeconomic factors on the association between combinations of unhealthy lifestyle factors and adverse health outcomes.

► The inclusive nature of the eligibility criteria, which is necessary as there are likely to be few studies in this area, means included studies may be heterogeneous in design and methodology and this may preclude meta-analysis.

► The wide range of possible socioeconomic indicators and combinations of lifestyle factors likely to be included due to the inclusion criteria may make firm conclusions difficult.

► However, the wide range of possible socioeconomic indicators and combinations of lifestyle factors likely to be included due to the inclusion criteria will permit a comprehensive overview of both sets of exposures and, therefore, highlight evidence gaps.

► Synthesising a broad evidence base to provide an overview of the potential influence of socioeconomic status (SES) on associations between combinations of unhealthy lifestyle factors and adverse health outcomes could indicate which combinations of unhealthy lifestyle factors are associated with the greatest risks for lower SES groups.

## INTRODUCTION
### Background

Globally, unhealthy lifestyle factors (eg, smoking, physical inactivity) are among the main risk factors for mortality and non-communicable diseases (NCDs).[1] Socioeconomically deprived populations have the highest mortality and morbidity rates from NCDs but this is only partially explained by higher prevalence of unhealthy lifestyle factors.[2–4] Deeper explanations for

lifestyle-related health inequalities include both the synergistic interactions between individual lifestyle factors themselves and interactions between lifestyle factors and socioeconomic status (SES).[5 6] However, to date, there has been limited examination of how the association between combinations of unhealthy lifestyle factors and adverse health outcomes is moderated by SES.

It is critical to note that the word 'lifestyle' implies choice and control over health behaviours. However, lower SES (more deprived, lower income or less educated) groups are less likely to have as much control over 'lifestyle' or health behaviours than higher SES groups. Further, 'choosing' unhealthy health behaviours may be entirely rational given specific socioeconomic contexts.[7] However, the word lifestyle is employed here as this is widely understood in the context of potentially modifiable health behaviours.

## Combinations of lifestyle factors

While single lifestyle factors are clearly associated with adverse health outcomes, meta-analyses provide evidence for how combinations of lifestyle factors have stronger associations with mortality and NCDs.[5 8 9] The evidence for the health impact of single lifestyle factors now also includes 'new' or emerging lifestyle factors, such as sleep duration,[10] television viewing time[11] and social participation levels.[12] When 'new' lifestyle factors are combined with 'conventional' factors (eg, smoking, physical inactivity, high alcohol intake or poor diet) associations with mortality are strengthened further.[12] Moreover, combinations of lifestyle factors can have a multiplicative or synergistic effect on adverse health outcomes. For example, the mortality associated with smoking and high alcohol intake together is more than the sum of the risks for each lifestyle factor alone.[13] Therefore, investigating the impact of broad combinations of lifestyle factors is necessary for comprehensive understanding of lifestyle-associated harm. Particularly so when the prevalence of three or more lifestyle risk factors is high.[14] For example, 55% of the Scottish population has three or more risk factors.[15] Furthermore, the additional risks associated with combinations of unhealthy lifestyle factors would motivate work to determine which combinations have the highest risk. For example, if a combination of high sedentary time together with short sleep duration and poor diet is highlighted as particularly high risk then interventions could be targeted at this specific behavioural combination.

## SES and lifestyle

There is a significant body of research that focusses on investigating the extent to which the greater prevalence and nature of unhealthy lifestyle factors in lower SES populations can explain the well-known socioeconomic gradient in adverse health—so-called 'differential exposure'.[3 4 16–21] These studies estimate that 30%–50% of socioeconomic inequalities in all-cause and cause-specific mortality are attributable to the differential exposure to unhealthy lifestyle factors. Typically, these studies examine conventional lifestyle factors only, although often alongside metabolic factors such as blood pressure or body mass index (BMI).

However, despite being independently associated with mortality and NCD at levels commensurate with those of unhealthy lifestyle factors, socioeconomic factors are often omitted from lifestyle policy.[22] Furthermore, many studies appear to lack an assessment of the interaction between unhealthy lifestyle factors and SES. There is some evidence for interactions between single lifestyle factors and SES, whereby, for the same level of exposure, lifestyle factors have different effects across socioeconomic spectra—that is, 'differential vulnerability'.[21] For example, in a Scottish cohort, lower (as opposed to higher) SES (measured by education level, social class, household income and area-based deprivation) had stronger associations with alcohol-related hospital admissions and alcohol-related deaths at the same level of alcohol intake even after controlling for drinking patterns, smoking and BMI.[23] Similarly, excess harm in lower SES groups has been associated with the single lifestyle factors of smoking and physical inactivity.[24] The underlying mechanisms that explain differential vulnerability remain unclear but could include interactions between lower SES and other harmful factors associated with low SES (eg, other unhealthy lifestyle factors, stress, reduced access to healthcare) or be due to accelerated biological ageing in lower SES groups due to greater cumulative life course risks (eg, increased frequency of adverse childhood experiences, poorer childhood health).[25 26] However, differential vulnerability shown in these observational studies may also represent an artefact of residual confounding or could be due to lack of detail in survey or interview measurements of lifestyle factors which fails to fully capture greater intensiveness (differential exposure) of unhealthy lifestyle factors in lower SES groups (eg, lower SES groups who drink heavily may drink more than heavy drinkers in higher SES groups).[27]

More recently, there has been investigation of the interaction between combinations of lifestyle factors and SES.[3 6 21 28 29] Some studies show lower SES being associated with disproportionately higher cardiovascular disease (CVD) and all-cause mortality with combinations of unhealthy lifestyle factors.[6 30] Examining the evidence for SES influence on adverse health associated with combinations of unhealthy lifestyle factors would help unpack the evidence for and against differential vulnerability and improve our understanding of wider lifestyle associated risks across SES spectra. However, to our knowledge, there has been no systematic review of the evidence for interactions between SES and combinations of unhealthy lifestyle factors in terms of adverse health outcomes. This paper describes the protocol for a systematic review of the effects of SES on the association between combinations of unhealthy lifestyle factors and adverse health outcomes. This review will highlight evidence gaps and deepen our understanding of the complex interplay between lifestyle, SES and adverse health outcomes. Findings from this review will inform the development

**Table 1** PICOS inclusion and exclusion criteria

| PICOS element | Description |
|---|---|
| Population | Studies of any general population type will be included. Eligibility will not be restricted by age, sex, or other sociodemographic characteristics. Cohort studies focusing on participants with an index condition/disease will be excluded. |
| Exposure | Studies that examine two main exposures of interest will be included:<br>1. Lifestyle factor combinations<br>Combinations must include at least three lifestyle factors and may include any combination of either conventional or emerging lifestyle factors. Combinations may include metabolic or intermediate risk factors (eg, blood pressure, cholesterol, or body mass index) but at least three factors included in the combination must be behavioural lifestyle factors (eg, smoking, physical activity and diet) as opposed to intermediate or metabolic factors.<br>2. Socioeconomic status<br>All SES measures will be permitted. Anticipated variables include but are not limited to individual or area-based measures of education, employment, occupation, income and deprivation or poverty indices. |
| Comparator | Studies will be included where reported findings allow an assessment of the impact of SES on the association between combinations of lifestyle factors and adverse health outcomes.<br>Comparisons of effects for available outcomes will be made, for example, HRs of participants with the 'unhealthiest' lifestyle factor combination in the most affluent SES group will be compared with the HRs of participants with the unhealthiest lifestyle but in the least affluent SES group (ie, unhealthy + high SES vs unhealthy + low SES).<br>We will compare results for tests of interaction between lifestyle factor combinations and SES measures. |
| Outcomes | Primary outcome:<br>► All-cause mortality<br>Secondary outcomes:<br>► CVD and cancer mortality<br>► CVD and cancer incidence |
| Study | Prospective observational cohort studies.<br>Studies published in English language. |
| Exclusions | Ineligible publication/study design (eg, reviews, conference abstracts, case–control and cross-sectional studies, intervention studies, qualitative studies).<br>Studies lacking exposures or outcomes of interest (eg, combinations of fewer than three lifestyle factor or SES not examined).<br>Studies that do not provide an assessment of the impact of SES on the association between combinations of lifestyle factors and adverse health. |

CVD, cardiovascular disease; PICOS, population, intervention, comparator, outcome, study design; SES, socioeconomic status.

of policy and research that aims to better support and understand healthy lifestyles and contribute to reducing the excess lifestyle-related mortality and morbidity in lower SES populations.

## Aims

This review aims to identify, appraise and synthesise the findings from studies that examine the effects of SES on the association between combinations of unhealthy lifestyle factors and adverse health outcomes. This review has two key questions:

1. What are the characteristics of studies that examine the effect of SES on the association between combinations of unhealthy lifestyle factors and adverse health outcomes?
2. What is the evidence for whether and how the association between combinations of unhealthy lifestyle factors and adverse health outcomes is moderated by SES?

## METHODS AND ANALYSIS

This systematic review is registered with the international database of prospectively registered systematic reviews.[31]

## Eligibility criteria

Inclusion criteria are presented in table 1 according to an adapted population intervention, comparator, outcome, study design framework from the Cochrane Handbook, where 'I' (intervention) is replaced with 'E' (exposure).[32]

### Population

This review will focus on the impact of SES on lifestyle associated adverse health outcomes in the general population. Of the studies included in previous systematic reviews investigating the adverse health outcomes associated with combinations of lifestyle factors, very few included an evaluation of the impact of SES.[5 8 9] Therefore, because it was anticipated that few studies would fit the inclusion criteria, the population type was not restricted in order to identify as many studies as possible.

### Exposure

Only studies that examine both combinations of lifestyle factors and SES as exposure variables will be included. Studies that examine the combined influence of at least three lifestyle factors will be included. It was decided that three lifestyle factors represented a balance between identifying the evidence for combinations of lifestyle factors as opposed to single lifestyle factors (two lifestyle factors

**Table 2** PubMed search strategy

| Search | MeSH terms and keywords | Theme |
|---|---|---|
| #1 | combination*[tiab] OR combined[tiab] OR composite[tiab] OR integrated[tiab] OR interaction*[tiab] OR joint effect* OR merged effect*[tiab] OR score*[tiab] OR adhere* to[tiab] OR collective[tiab] OR cumulative[tiab] OR multiple[tiab] | combined |
| #2 | life style[MeSH] OR life style*[tiab] OR lifestyle*[tiab] OR risk reduction behavior[MeSH] OR risk reduction behaviour*[tiab] OR health behavior[MeSH] OR health behaviour[tiab] OR health factor*[tiab] OR low risk*[tiab] OR prevention guideline*[tiab] OR protective factor*[tiab] OR risk reduction behaviour*[tiab] OR health* behaviour*[tiab] OR risk behaviour*[tiab] OR modifiable factors[tiab] | lifestyle factors |
| #3 | healthcare disparities[MeSH] OR healthcare disparities[tiab] OR Health Status Disparities[MeSH] OR disparate[tiab] OR disparit*[tiab] OR inequal*[tiab] OR health inequalities[tiab] OR unequal[tiab] OR health inequities[tiab] OR inequit*[tiab] OR socioeconomic factors[MeSH] OR socioeconomic factors[tiab] OR socio-economic*[tiab] OR socioeconomic*[tiab] OR social-economic[tiab] OR Social Determinants of Health[MeSH] OR social determinant*[tiab] OR poverty[MeSH] OR poverty[tiab] OR depriv*[tiab] OR sociological factors[MeSH] OR sociological factors[tiab] OR social medicine[MeSH] OR social medicine[tiab] | SES |
| #4 | cohort studies[MeSH] OR cohort[tiab] OR incidence[MeSH] OR incidence[tiab] OR survival analysis[MeSH] OR survival[tiab] OR early diagnosis[MeSH] OR early diagnosis[tiab] OR prospective*[tiab] OR follow* up[tiab] OR longitudinal[tiab] OR nested case-control[tiab] OR nested case control[tiab] OR predict*[tiab] | study design |
| #5 | #1 AND #2 AND #3 AND #4 | - |
| #6 | Death[MeSH] OR death*[tiab] OR mortality[MeSH] OR mortalit*[tiab] OR fatal*[tiab] OR life expectanc*[tiab] OR surviv*[tiab] | mortality outcome |
| #7 | cardiovascular diseases[MeSH] OR cardiovascular[tiab] OR CVD[tiab] OR heart disease*[tiab] OR myocardial ischaemia[tiab] OR AMI[tiab] OR IHD[tiab] OR CHD[tiab] OR coronary artery disease*[tiab] OR CAD[tiab] OR myocardial infarction[tiab] OR heart infarction[tiab] OR acute coronary syndrome[tiab] OR ACS[tiab] OR heart failure[tiab] OR sudden cardiac death[tiab] OR cerebrovascular disorder*[tiab] OR cerebrovascular accident*[tiab] OR cerebrovascular attack*[tiab] OR CVA[tiab] OR cerebrovascular disease*[tiab] OR CBVD[tiab] OR cerebral arterial disease*[tiab] OR stroke*[tiab] OR apoplex*[tiab] | CVD outcomes |
| #8 | neoplasms[MeSH] OR neoplas*[tiab] OR cancer*[tiab] OR carcinoma*[tiab] OR tumour*[tiab] OR malignanc*[tiab] | cancer outcomes |
| #9 | #6 OR #7 OR #8 | - |
| #10 | #9 AND #5 (final search) | - |

(MeSH)=Medical Subject Heading; (tiab)=contained in either title or abstract; underlined=both UK and American spellings will be searched; *=any group of letters/characters, including no character.
CVD, cardiovascular disease; SES, socioeconomic status.

was felt to be too narrow) while ensuring that a sufficient number of studies are included (there are fewer studies examining the risks of >3 lifestyle factors). In order to help identify as much literature as possible we decided that all definitions of SES variables will be accepted.

## Comparator

Studies will be included if they examine the effect of SES on the associations between combinations of lifestyle factors and adverse health outcomes. Results for effects may be reported in different ways: HRs, ORs, incidence rates. Where possible, comparisons of effects for similar outcomes will be made across studies. For example, the HRs of participants with the least healthy lifestyle factor combination in the most affluent SES group will be compared with the HRs of participants with the least healthy lifestyle combination but in the least affluent SES group. Where reported, we will compare results for tests of interaction between lifestyle combinations and SES measures.

## Outcomes

Our primary outcome of interest is all-cause mortality. However, lifestyle-associated adverse health is well recognised to be strongly linked to CVD and cancer outcomes. Therefore, we will include studies that examine the following outcomes: CVD and cancer mortality; CVD and cancer incidence. Studies examining specific CVD or cancer outcomes such as stroke, angina or site-specific cancer will also be included. The International Classification of Diseases (10th revision) codes I05–I89.9 and C00-C97 will be used to define CVD and cancer outcomes, respectively.

## Study design

We aim to identify prospective observational cohort studies. Case–control and cross-sectional studies, intervention studies, qualitative work and review articles will be excluded. Only full-text published articles will be included and conference abstracts, dissertations,

editorials or papers without data will be excluded. Studies not published in the English language will be excluded.

## Study identification
### Electronic searches
A systematic search of PubMed, EMBASE and CINAHL databases will be performed. The search strategy will incorporate a combination of Medical Subject Heading (MeSH) terms and keywords. The search strategy of a recent systematic review examining combined lifestyle factors and the risk of incident type 2 diabetes was used as a template and adapted to incorporate SES related MeSH terms and keywords.[8] The search strategy has been developed with assistance from a specialist university librarian. Table 2 shows the search strategy that will be used for PubMed. The search strategy will be adapted and applied to other databases and will be available from PROSPERO once the review is complete.

Searches will be from database inception (PubMed 1966; EMBASE 1947; CINAHL 1984) to 3 March 2020. Searches will be supplemented by handsearching of reference lists of included papers, forward citation searching and a search for grey literature using the following sources:

- ► Charities/health organisations: for example, The King's fund, The Health Foundation, Cancer Research UK, WHO, American Heart Association, American Cancer Society.
- ► Databases such as OpenGrey, the Healthcare Management Information Consortium, the National Technical Information Service.
- ► Google and Google Scholar.
- ► Literature compiled by governmental organisations for example, Department of Health in England, Office for National Statistics, Centers for Disease Control and Prevention.

## Data collection and analysis
### Study selection
Studies identified by the search strategy will be uploaded to 'DistillerSR' software and duplicates will be removed. Two reviewers will independently screen titles and abstracts using the inclusion criteria above. Any conflicts shall be resolved by discussion and if an agreement cannot be made the study shall be included for full-text screening.

Full texts will be reviewed using a piloted checklist based on the inclusion and exclusion criteria. Conflicts will be resolved by discussion and will include a third reviewer if no consensus is reached. All studies excluded at the full-text stage will be listed with reasons for exclusion given.

### Data extraction
Each study that meets inclusion criteria after full-text screening will go through the data extraction phase. Data extraction will be carried out by two reviewers working independently using a piloted data extraction form (box 1). Data will be extracted for the following

study characteristics: author, publication year, title, study cohort, number of participants, proportion female, mean age, ethnicity, setting, country, date of recruitment, duration/follow-up. Details of the lifestyle or metabolic factors and SES measures used as exposure variables will be extracted. Where possible, this detail will include how and when exposure variables were measured or assessed. Data will be extracted for any included metabolic factors such as BMI, blood pressure or cholesterol levels. The number of study participants with unhealthy lifestyle factors will be recorded and reported. Details of confounder variables and the level of missing data will be extracted. Health outcome definitions and ascertainment will be recorded. The type of analysis, statistical methodology and confounder adjustment will be extracted. Study results, the nature of adverse health outcome associations identified and the effect sizes that measure the impact of SES on lifestyle associated outcomes will be recorded.

---

### Box 1  Data extraction

**Article identifiers**
- ► Author.
- ► Publication year.
- ► Title.
- ► Journal, Vol, Issue, Page numbers.
- ► Source (eg, Database, Grey Literature source, handsearching of references, etc.).
- ► Study funding.

**Study characteristics**
- ► Study aims and objectives.
- ► Cohort name.
- ► Number of participants.
- ► Proportion female.
- ► Mean age (SD).
- ► Ethnicity.
- ► Country.
- ► Setting (eg, general population, occupational cohort, etc.)
- ► Study duration/follow-up.
- ► Study start and end dates.

**Exposures, confounders and outcomes**
- ► Lifestyle and/or metabolic factors (definition; when and how measured/assessed).
- ► SES measures (definition; when and how measured/assessed).
- ► Confounder variables or covariates included.
- ► Outcome definition.
- ► Outcome ascertainment.
- ► Number of participants with missing data.

**Analysis characteristics**
- ► Type of analysis (statistical methods).
- ► Sensitivity analysis conducted.
- ► Methods to deal with missing data.

**Results, conclusions, and quality**
- ► Effect of SES on lifestyle-associated adverse health: yes or no.
- ► Size of effect.
- ► Study conclusions.
- ► Newcastle-Ottawa Scale and justification.

---

Box 2    Adapted Newcastle-Ottawa scale†

**Selection**
1. Representativeness of the cohort
   – Truly representative of those about whom the study makes conclusions (adult/general/low income population).*
   – Somewhat representative.*
   – Selected group of users, for example, nurses, volunteers.
   – No description of the derivation of the cohort.
2. Ascertainment of exposures (lifestyle factors and SES measures)
   – Secure record (eg, healthcare records).*
   – Structured interview.*
   – Written self-report.
   – No description.
3. Analysis rigour
   – Impact of >1 socioeconomic measure assessed.*
   – Impact of one socioeconomic measure assessed.
4. Demonstration that participants were disease free at start of study
   – Yes/sensitivity analysis excluding those with prior disease.*
   – No.

**Comparability (two stars available)**
1. Comparability of exposed versus unexposed on the basis of the design or analysis
   – Analysis includes adjustment for key variables and maximises the use of available data.*
   – Study offers reasonable justification for adjustment variables.*

**Outcome**
1. Assessment of outcome
   – Independent blind assessment.*
   – Record linkage.*
   – Self-report.
   – No description.
2. Was follow-up long enough for outcomes to occur
   – Yes (≥6 months).*
   – No.
3. Adequacy of follow-up of cohorts
   – All participants accounted for and missingness <50%, or assessment of impact of missingness given, or description of those with missing data allows assessment of impact of missingness.*
   – Some participants unaccounted for or missingness >50%, or no assessment of impact of missingness, or insufficient description of those with missing data prevents assessment of impact of missingness.

* = star awarded.
†A study can be awarded a maximum of one star for each numbered item within the selection and outcome categories. A maximum of two stars can be given for comparability. A total of nine stars are available: selection (four stars), comparability (two stars) and outcome (three stars). If a single study combines/harmonises multiple cohorts for analysis, then the item that best fits with the majority of the cohorts included should be given for that study.

Whether and how sensitivity analyses were conducted will be noted. Techniques for dealing with missing data will be recorded. Studies' overall conclusions will be extracted.

### Quality

Included studies will be assessed for quality using an adapted version of the Newcastle-Ottawa Scale for cohort studies,[33] a tool that has been used extensively for the appraisal of observational studies as described here. This scale, after piloting, has been adapted to include an assessment of confounder adjustment, sensitivity analyses, and dealing with missing data (box 2).

### Data synthesis

The process and results of study identification and selection based on inclusion and exclusion criteria will be displayed as a Preferred Reporting Items for Systematic Reviews and Meta-Analyses flow diagram.[34]

To aid comparisons across studies, we will present at least the following summary data for each included study in tabular format:
► Cohort characteristics (eg, number of participants, proportion female, mean age, setting, date of recruitment, length of follow-up).
► Lifestyle or metabolic factors included in combination.
► SES measures used in assessment of SES effects
► Health outcomes and outcome ascertainment.
► Risks for health outcomes and their statistical significance.
► Inconsistent findings within each study.
► Study quality.

Studies will be grouped together according to our outcomes of interest (all-cause mortality; CVD and cancer mortality; CVD and cancer incidence). For all outcomes, we will describe details of how the outcome was assessed (eg, administrative data or questionnaires) and approach used for analysis (eg, time-to-event). We will provide details of the association between SES, lifestyle and the outcome (eg, OR, HR, etc) including the length of follow-up. We will report studies attempts to deal with confounding and discuss whether resulting associations are likely confounded. Where possible, comparisons of effects for similar outcomes will be made across studies. For example, we will compare HRs of participants in the least healthy category and most affluent SES category (reference group) with HRs of participants in the least healthy category but in the most deprived SES category.

If studies are sufficiently homogeneous in terms of participant characteristics, exposures and outcomes, we will standardise study findings for similar outcomes, provide justification for our transformation methods and combine results by a random-effects meta-analysis.[35] We will then calculate $I^2$ to describe the proportion of effect estimate variance due to study heterogeneity rather than chance.

Initial scoping of the literature has identified significant exposure, outcome and methodological heterogeneity across studies. Therefore, a synthesis without meta-analysis (SWiM) will likely be the most appropriate method to synthesise study findings.[36] As per SWiM, we will provide justifications for the method and presentation of study findings.

Irrespective of whether a meta-analysis is conducted, we will provide a transparent and full account of any limitations of our synthesis. Further, in conducting and reporting this systematic review, we will endeavour to fulfil,

where possible, all items proposed by the Meta-analysis Of Observational Studies in Epidemiology (MOOSE) Group.[37] Any amendments to the review protocol will be identified and justified on completion.

## Patient and public involvement

This Systematic Review constitutes a primary aspect of HF's doctoral thesis. National Health Service Research Scotland Primary Care Patient and Public Involvement (NRS PPI) Group was consulted twice as part of preparatory work for the doctoral thesis funding application.[38] The NRS PPI Group were not involved in the design of the study but have influenced how results of this review, as well as other aspects of the doctoral thesis, will be presented at two planned public engagement events over the course of the thesis.

## ETHICS AND DISSEMINATION

This review will not require ethical approval as it will not involve individual-level patient data. Results will be disseminated via peer-reviewed publication, professional networks, social media, public events and conference presentations.

## DISCUSSION

Both combinations of multiple unhealthy lifestyle factors and SES play major roles in mortality, CVD and cancer.[1 5 8 9 22] Numerous studies have investigated the mediating influence of lifestyle factors in attempts to explain the socioeconomic gradient in adverse health outcomes.[3 4] However, fewer studies appear to examine the relationships between lifestyle, SES and adverse health outcomes from the perspective of interactions between combinations of lifestyle factors and SES.[28] Understanding the evidence for whether and how SES influences the association between lifestyle and adverse health outcomes could inform policies and interventions that aim to support healthy lifestyles.

Scoping the literature suggests that evidence for a moderating influence of SES is mixed. Eguchi et al[28], with data from 42 647 Japanese adults aged 40–79 years and approximately 20 years follow-up, examined the risks associated with a lifestyle score (comprised of eight lifestyle factors: smoking, alcohol, physical activity, sleep, dietary intake of fruit, fish and milk, and BMI) stratified by age (≥16 or <16 years) at last formal education. The authors reported a 44% higher all-cause mortality risk for those with a higher level of education but with the least healthy lifestyle, compared with those with a higher level of education but with the healthiest lifestyle. When the same comparison was made in those with the lower level of education, participants with the least healthy lifestyle had a 40% higher all-cause mortality risk compared with those with the healthiest lifestyle. Namely, the level of elevated risk associated with the least healthy lifestyle was similar in both higher and lower education groups.

Foster et al[6], performed similar analyses with data from 328 594 UK adults 40–69 years and approximately 5 years follow-up to examine the risks associated with a lifestyle score (comprised of nine lifestyle factors: smoking, alcohol, physical activity, television viewing time, sleep duration, and dietary intake of fruit/vegetables, oily fish, and red and processed meat) stratified by quintiles of socioeconomic deprivation (Townsend index). The authors observed a 65% higher all-cause mortality risk for the least deprived with the least healthy lifestyle, compared with the least deprived but with the healthiest lifestyle. However, when the same comparison was made in the most deprived participants, those with the least healthy lifestyle had a 145% higher risk than those with the healthiest lifestyles.

These highly comparable studies report opposing results. Eguchi et al[28] found no interaction between lifestyle and SES with similar lifestyle risks in the least and most educated groups. Whereas Foster et al[6] found an interaction between lifestyle and SES with disproportionately raised lifestyle risk in the most deprived group. Several methodological differences (lifestyle and SES measures; follow-up time; population characteristics) could explain the two studies' conflicting results but additional evidence from similar studies would help to clarify whether there is any moderating influence of SES on lifestyle-associated risks. This systematic review will help unpack such associations in more detail. In addition, included studies may identify specific combinations of unhealthy lifestyle factors that pose the highest risks for lower SES groups. However, we suspect there is likely to be a lack of studies which identify the combinations that pose the greatest risk for lower SES groups and this may be one of the evidence gaps that this review identifies.

To our knowledge, this systematic review will be the first to synthesise the evidence for whether and how SES influences the association between combinations of lifestyle factors and adverse health. We will describe the lifestyle factors, SES measures and adverse health outcomes that have been examined thus far. We will collate and interpret the findings considering both the type of analyses and the quality of studies to provide a comprehensive synthesis of available evidence and highlight gaps in current evidence.

We have developed a comprehensive search strategy with broad inclusion criteria in order to identify all available evidence and reduce the chance of omitting relevant studies. However, our scoping of the literature suggests that not only will there be few studies that attempt to examine this problem but also that previous studies will be widely heterogeneous both in terms of the lifestyle factor and SES variables examined and in terms of the statistical methods employed. This will likely preclude a meta-analytical synthesis of the evidence, which may be a limitation of our review. However, SWiM will likely highlight important gaps in available evidence and direct future research in this sphere. This review will adhere to SWiM reporting guidelines and will be guided by MOOSE recommendations to improve transparency and clarity.[36 37]

All screening, data extraction and quality assessment will be performed independently by two reviewers to improve study rigour.

This comprehensive and rigorous systematic review will improve our understanding of the complex interaction between SES and lifestyle and has the potential to inform research, interventions and policy.

**Contributors** HF, CAO, FM and JG were involved in study concept. HF, CAO, FM, PP and JG developed the study design. HF, CAO, FM, JG and PP were involved in acquisition, analysis or interpretation of data. Drafting of the manuscript was led by HF and PP with supervision and support from CAO, FM and JG. All authors were involved in critical revision of the manuscript for important intellectual content. CAO is the guarantor.

**Funding** HF is funded by a Medical Research Council Clinical Research Training Fellowship (MRC CRTF; MR/T001585/1) and this body of work constitutes a primary aspect of HF's doctoral thesis.

**Disclaimer** The funder had no role in developing this protocol.

**Competing interests** None declared.

**Patient and public involvement** Patients and/or the public were involved in the design, or conduct, or reporting, or dissemination plans of this research. Refer to the Methods section for further details.

**Patient consent for publication** Not required.

**Provenance and peer review** Not commissioned; externally peer reviewed.

**ORCID iDs**
Hamish Foster http://orcid.org/0000-0002-0224-7125
Peter Polz http://orcid.org/0000-0003-1524-8482
Catherine A O'Donnell http://orcid.org/0000-0002-5368-3779

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
