## [Reviewer comments · BMJ Open]

ARTICLE DETAILS

TITLE (PROVISIONAL)	Understanding the influence of socioeconomic status on the association between combinations of lifestyle factors and adverse health outcomes: a systematic review protocol.
AUTHORS	Foster, Hamish; Polz, Peter; Mair, Frances; Gill, Jason; O'Donnell, Catherine

VERSION 1 – REVIEW

REVIEWER	David Bann UCL
REVIEW RETURNED	23-Sep-2020

GENERAL COMMENTS	I think that this is broadly speaking a well-written and interesting protocol. Some suggestions below: • The word 'lifestyle' has unfortunate connotations, particularly in this context. It is suggestive of a style of life, in which individuals willingly and mistakenly choose. Individuals have less 'choice' to choose their 'lifestyle' than is commonly believed, as the social determinants of health literature repeatedly suggests. It would be unfortunate for this work to be judged negatively due to this simply phrasing. I suggest this is re-considered (eg, health-impacting behaviours, health behaviours, behaviours)• Abstract, introduction: would be helpful to note why the moderation would occur.• "However, socioeconomic factors themselves exert an independent effect on health and SES should be considered a risk factor alongside lifestyle factors.(21)" I suggest re-wording/re-thinking this section. It is a risk factor by definition.• It would be helpful to be clear throughout about why there could be modification. Both real mechanisms (eg, differential vulnerability as the authors note), but potentially also artefactual (eg, if the different SES groups are non-comparable in confounding factors)• If there is a clear set of studies which examine single behaviours and moderation by SES, this would seem useful to include. Explaining the effect of adding multiple behaviours together will ultimately need to be explained in terms of the individual behaviours. It wasn't clear if these studies were reviewed here or elsewhere.• May be useful to specify a-priori the behaviours included for selection. The paper search strategy could also use some explicitly, and not use terms such as lifestyle. Important to check this to ensure key papers are not missed.• Is it possible that such associations are confounded? This doesn't appear to feature in box 2. Studies may not focus overly on the moderation aspect, so may not include important
---

	confounders of the moderation component of the association. Perhaps factors like age (eg, wealth differences are substantial by age), preceding illness etc.
--	--

REVIEWER	Akira Shibamura The University of Tokyo, Japan
REVIEW RETURNED	09-Nov-2020

GENERAL COMMENTS	This study protocol explains the systematic review and possible meta-analysis study regarding the mediating roles of socio-economic status in the association between unhealthy lifestyle factors and adverse health outcomes. I agree with the authors on the importance of this study. This study possibly highlights how people in deprived social groups had unfavorable conditions in their health status when they are exposed to unhealthy lifestyle. I would like the authors to clarify several points to be pointed below.  1. What if there are studies that investigated unhealthy lifestyle as a mediator in the association between SES and health outcome and that reported the difference in the risk of health outcomes by high and low SES when participants had unhealthy lifestyle. Do the authors include these studies? Please clarify. 2. Both SES and the exposure to unhealthy lifestyle can change overtime during the study period. Can prospective cohort studies capture the interrelationship among SES, unhealthy lifestyle, and health outcomes? Under a conventional understanding of prospective cohort study design, exposure variable may be good to be time-invariant. Please discuss. 3. It is highly likely that combining multiple lifestyle factors cause adverse health outcomes, compared with a single lifestyle factor, as the authors explained in the Background section. While the authors review articles with multiple unhealthy lifestyle factors as exposure, they may not be able to confirm how multiple unhealthy lifestyle factors accelerate adverse health outcomes, compared with a single unhealthy lifestyle factor. The authors may want to note that there is a risk of exaggerating the finding since a particular lifestyle factor among the combined factors could largely cause adverse health outcomes. 4. Related to the comment above, I believe that this review will be able to report the proportion of study participants who have each of unhealthy lifestyle factors in addition to the proportion of combined unhealthy lifestyle factors. The information can give readers better understanding of unhealthy lifestyle factors that study participants have in its study site. 5. How do studies included in this review possibly have different statistical methods? The analysis of mediating factors and interaction term analysis have different statistical approach. And the reporting and possible synthesis may be more complicated than typical systematic reviews. The authors may be able to explain different statistical methods for analyzing mediating factors and how synthesis might or might not be possible. 6. In the Background and Methods sections, the authors identified unhealthy lifestyle factors (such as, smoking, physical inactivity, alcohol intake, poor diet, sleep duration, television viewing, and social participation). Are there reasons why they did not use these factors as search terms for unhealthy lifestyle factors? There may have pros and cons of clarifying the fixed terms for lifestyle factors (for example, synthesis could be easier if fixed terms are used but the review would fail to include studies that did not use such fixed
--

	terms). The authors may be able to explain and justify their approach.
--	--

REVIEWER	Dr Anne Eaton Memorial Sloan-Kettering Cancer Center, Biostatistics
REVIEW RETURNED	10-Nov-2020

GENERAL COMMENTS	You are looking for studies that examine “lifestyle factor combinations” that include at least 3 lifestyle factors. It wasn’t clear to me whether the combinations had to be represented as a single “score”, or whether you would include a study that looked at the simultaneous effect of several lifestyle factors in a multivariable regression model. This sentence is not clear. Stronger than what?: “For example, in a Scottish cohort, lower SES (measured by education level, social class, household income, and area-based deprivation) had stronger associations with alcohol-related hospital admissions and alcohol related-deaths at the same level of alcohol intake even after controlling for drinking patterns, smoking and BMI.” In Table 1, it seems that you saying that to be included, a study must not restrict participants based on age, sex, etc. This seems to contradict what you say later: “Therefore, because it was anticipated that few studies would fit the inclusion criteria, the population type was not restricted in order to identify as many studies as possible.” In table 2, I don’t know what #1, #2, etc mean. Is #10 is the final search you will use? A potential limitation that should be mentioned is that different lifestyle factor combinations may represent totally different things, which will make it hard to draw conclusions. There is no reason to expect that the presence or absence (or direction) of an interaction between SES and a lifestyle factor combination would be the same for different lifestyle factors or lifestyle factor combinations, especially very different ones like smoking versus social participation.
--

VERSION 1 – AUTHOR RESPONSE

Reviewer: 1

Comments to the Author

I think that this is broadly speaking a well-written and interesting protocol. Some suggestions below:
(Authors’ response) We thank the reviewer for their positive comment.

1. The word ‘lifestyle’ has unfortunate connotations, particularly in this context. It is suggestive of a style of life, in which individuals willingly and mistakenly choose. Individuals have less ‘choice’ to choose their ‘lifestyle’ than is commonly believed, as the social determinants of health literature repeatedly suggests. It would be unfortunate for this work to be judged negatively due to this simply phrasing. I suggest this is re-considered (eg, health-impacting behaviours, health behaviours, behaviours)

(Authors’ response) We thank the reviewer for this helpful reminder. We too would not want our paper to be judged negatively because of our use of the word ‘lifestyle’. We entirely agree that the word ‘lifestyle’ has unhelpful connotations, and this is of particular importance when considering wider

socioeconomic influences where the opportunity for healthy living is not evenly distributed through populations.

Therefore, we have added the following at the start of the paper to acknowledge the problem with the word 'lifestyle' (line 14):

"It is critical to note that the word 'lifestyle' implies choice and control over health behaviours.

However, lower SES (more deprived, lower income, or less educated) groups are less likely to have as much control over 'lifestyle' or health behaviours than higher SES groups. Further, 'choosing' unhealthy health behaviours may be entirely rational given specific socioeconomic contexts.(7)

However, the word lifestyle is employed here as this is widely understood in the context of potentially modifiable health behaviours."

2. Abstract, introduction: would be helpful to note why the moderation would occur.

(Authors' response) We are grateful for the reviewer for highlighting this and we have added the following to the abstract, introduction:

"Lower SES groups may be disproportionately vulnerable to the adverse effects of unhealthy lifestyle factors compared to higher SES groups via interactions with other factors associated with low SES (e.g., stress, reduced access to health care) or via accelerated biological ageing."

3. "However, socioeconomic factors themselves exert an independent effect on health and SES should be considered a risk factor alongside lifestyle factors.(21)" I suggest re-wording/re-thinking this section. It is a risk factor by definition.

(Authors' response) We apologise for the lack of clarity here. The reference here (Stringhini et al. Lancet. 2017;389(10075):1194-) highlights how the WHO 25x25 initiative does not include socioeconomic status as a risk factor when considering modifiable NCD and mortality risks. We have altered the relevant section to now read (line 51):

"However, despite being independently associated with mortality and non-communicable disease at levels commensurate with those of unhealthy lifestyle factors, socioeconomic factors are often omitted from lifestyle policy.(22) Furthermore, many studies appear to lack an assessment of the interaction between unhealthy lifestyle factors and SES."

4. It would be helpful to be clear throughout about why there could be modification. Both real mechanisms (eg, differential vulnerability as the authors note), but potentially also artefactual (eg, if the different SES groups are non-comparable in confounding factors)

(Authors' response) We thank the reviewer, and we agree that adding information on potential mechanisms for SES modification improves the manuscript. Therefore, we have added the following to the introduction (line 62):

"The underlying mechanisms that explain differential vulnerability remain unclear but could include interactions between lower SES and other harmful factors associated with low SES (e.g., other unhealthy lifestyle factors, stress, reduced access to health care) or be due to accelerated biological ageing in lower SES groups due to greater cumulative life-course risks (e.g., increased frequency of adverse childhood experiences, poorer childhood health). However, differential vulnerability shown in these observational studies may also represent an artefact of residual confounding or could be due to lack of detail in survey or interview measurements of lifestyle factors which fails to fully capture greater intensiveness (differential exposure) of unhealthy lifestyle factors in lower SES groups (e.g., lower SES groups who drink heavily may drink more than heavy drinkers in higher SES groups)."

And (line 75):

"Some studies show lower SES being associated with disproportionately higher CVD and all-cause mortality with combinations of unhealthy lifestyle factors. Examining the evidence for SES influence on adverse health associated with combinations of unhealthy lifestyle factors would help unpack the evidence for and against differential vulnerability and improve our understanding of wider lifestyle associated risks across SES spectra."

5. If there is a clear set of studies which examine single behaviours and moderation by SES, this would seem useful to include. Explaining the effect of adding multiple behaviours together will ultimately need to be explained in terms of the individual behaviours. It wasn't clear if these studies were reviewed here or elsewhere.

(Authors' response) Studies which examine single unhealthy behaviours and moderation by SES are important. However, the aim of our work is to move on from examining single behaviours, which are seldom found in isolation, and consider the impact of multiple behaviours in combination, especially as we know that more socioeconomically deprived populations have a greater proportion of multiple unhealthy behaviours.

Our aim, therefore, is to examine the risks associated with wide combinations of unhealthy behaviours as we wish to capture broader range of health behaviour risks due to their potentially multiplicative risks and their clustering among lower SES groups. We have tried to make this clear in section entitled 'Combinations of Lifestyle Factors' and have added the following to this section to emphasise the point (line 35):

"Furthermore, the additional risks associated with combinations of unhealthy lifestyle factors would motivate work to determine which combinations have the highest risk. For example, if a combination of high sedentary time together with short sleep duration and poor diet is highlighted as particularly high risk then interventions could be targeted at this specific behavioural combination."

6. May be useful to specify a-priori the behaviours included for selection. The paper search strategy could also use some explicitly, and not use terms such as lifestyle. Important to check this to ensure key papers are not missed.

(Authors' response) We thank the reviewer for their comment but, as above, we are interested in a broad range of health behaviours in combination, including conventional (e.g. smoking and alcohol) and more recently appreciated health behaviours (e.g. sleep). As we are not focussing on individual or single health behaviours for this review, we have not included search terms for single health behaviours as we believe that any article that investigates combinations of health behaviours would likely include one of the combined/combo terms from our search strategy in the title or abstract.

7. Is it possible that such associations are confounded? This doesn't appear to feature in box 2. Studies may not focus overly on the moderation aspect, so may not include important confounders of the moderation component of the association. Perhaps factors like age (eg, wealth differences are substantial by age), preceding illness etc.

(Authors' response) We are grateful for this important reminder and agree that residual confounding will always be a potential explanation for results of observational studies. The 'Comparability' section of our adapted NOS scale has 2 stars available for assessing whether studies include adjustment for key variables (i.e. confounders) and whether studies offer reasonable justification for these adjustments. With regards preceding illness, number 4) in 'Selection' of NOS assesses whether studies demonstrate whether participants were 'disease free' at the start of the study as an attempt to deal with reverse causality. As part of our synthesis of the evidence we will discuss further whether any associations or results are likely to be due to residual confounding not controlled for in studies. As per our initial manuscript we state (line 203):

"Details of confounder variables and the level of missing data will be extracted. Health outcome definitions and ascertainment will be recorded. The type of analysis, statistical methodology, and confounder adjustment will be extracted."

We have added the following to state explicitly that the potential for confounding will be considered in our synthesis (line 278):

"We will report studies attempts to deal with confounding and discuss whether resulting associations are likely confounded."

Reviewer: 2

Comments to the Author

This study protocol explains the systematic review and possible meta-analysis study regarding the mediating roles of socio-economic status in the association between unhealthy lifestyle factors and adverse health outcomes. I agree with the authors on the importance of this study. This study possibly highlights how people in deprived social groups had unfavorable conditions in their health status when they are exposed to unhealthy lifestyle. I would like the authors to clarify several points to be pointed below.

(Authors' response) We thank the reviewer for their positive comment on the importance of this subject.

1. What if there are studies that investigated unhealthy lifestyle as a mediator in the association between SES and health outcome and that reported the difference in the risk of health outcomes by high and low SES when participants had unhealthy lifestyle. Do the authors include these studies? Please clarify.

(Authors' response) We are interested in how SES affects lifestyle associated risks and would include any study that provides data on those affects. Therefore, a study that examined the influence of lifestyle on the associations between SES and health outcomes would only be included if it also provided data on SES influences on the association between lifestyle and health outcomes.

As per our eligibility criteria (Table 1; Comparator): "Studies will be included where reported findings allow an assessment of the impact of SES on the association between combinations of lifestyle factors and adverse health outcomes."

The reason we are interested in this perspective rather than the perspective of how lifestyle affects SES related health outcomes is because the associations between unhealthy lifestyle factors and adverse health outcomes are well recognised but how those associations vary by SES level is less understood. We believe that evidence for SES differences in lifestyle-associated health would increase the imperative for addressing SES disparities.

2. Both SES and the exposure to unhealthy lifestyle can change overtime during the study period. Can prospective cohort studies capture the interrelationship among SES, unhealthy lifestyle, and health outcomes? Under a conventional understanding of prospective cohort study design, exposure variable may be good to be time-invariant. Please discuss.

(Authors' response) We thank the reviewer for their comment. We will be sure to report any limitation of the data, including the design of included studies. Some prospective cohorts collect data at multiple time points and if lifestyle and SES data are updated throughout follow-up this could reduce potential misclassification bias by capturing the change in participants' lifestyle or SES over time. Although cohorts with multiple data collection points would provide more accurate estimates for associations between lifestyle, SES, and health outcomes, we believe there are fewer of those cohorts and therefore, in order to capture as many studies as possible, we have not made this type of study design an inclusion criterion.

3. It is highly likely that combining multiple lifestyle factors cause adverse health outcomes, compared with a single lifestyle factor, as the authors explained in the Background section. While the authors review articles with multiple unhealthy lifestyle factors as exposure, they may not be able to confirm how multiple unhealthy lifestyle factors accelerate adverse health outcomes, compared with a single unhealthy lifestyle factor. The authors may want to note that there is a risk of exaggerating the finding since a particular lifestyle factor among the combined factors could largely cause adverse health outcomes.

(Authors' response) We thank the reviewer for their comment, which is very similar to the 5th comment of Reviewer 1. We have copied the response to their comment here. We hope this is a satisfactory response:

Studies which examine single unhealthy behaviours and moderation by SES are important. However, the aim of our work is to move on from examining single behaviours, which are seldom found in isolation, and consider the impact of multiple behaviours in combination, especially as we know that

more socioeconomically deprived populations have a greater proportion of multiple unhealthy behaviours.

Our aim, therefore, is to examine the risks associated with wide combinations of unhealthy behaviours as we wish to capture broader range of health behaviour risks due to their potentially multiplicative risks and their clustering among lower SES groups. We have tried to make this clear in section entitled 'Combinations of Lifestyle Factors' and have added the following to this section to emphasise the point (line 35):

"Furthermore, the additional risks associated with combinations of unhealthy lifestyle factors would motivate work to determine which combinations have the highest risk. For example, if a combination of high sedentary time together with short sleep duration and poor diet is highlighted as particularly high risk then interventions could be targeted at this specific behavioural combination."

4. Related to the comment above, I believe that this review will be able to report the proportion of study participants who have each of unhealthy lifestyle factors in addition to the proportion of combined unhealthy lifestyle factors. The information can give readers better understanding of unhealthy lifestyle factors that study participants have in its study site.

(Authors' response) We thank the reviewer for this idea and we agree that it would be helpful for readers to have this detail. Therefore, we have added this item to our data extraction and we will report this additional data in our systematic review. As a result, we have added the following line to 'Data Extraction' (line 203):

"The number of study participants with unhealthy lifestyle factors will be recorded and reported."

5. How do studies included in this review possibly have different statistical methods? The analysis of mediating factors and interaction term analysis have different statistical approach. And the reporting and possible synthesis may be more complicated than typical systematic reviews. The authors may be able to explain different statistical methods for analyzing mediating factors and how synthesis might or might not be possible.

(Authors' response) We agree that methodological differences will make comparisons difficult and may preclude meta-analysis. However, there are some methods that are more common than others and where methods are more similar between studies then synthesising results will be easier. However, as there is no consensus on the best way for assessing mediation or interactions, we will not limit our search strategy to only include specific methods. To acknowledge this difficulty, we have added a limitation in the 'Strengths and limitations of this study' section beneath the Abstract:
"•The inclusive nature of the eligibility criteria, which is necessary as there are likely to be few studies in this area, means included studies may be heterogenous in design and methodology and this may preclude meta-analysis."

6. In the Background and Methods sections, the authors identified unhealthy lifestyle factors (such as, smoking, physical inactivity, alcohol intake, poor diet, sleep duration, television viewing, and social participation). Are there reasons why they did not use these factors as search terms for unhealthy lifestyle factors? There may have pros and cons of clarifying the fixed terms for lifestyle factors (for example, synthesis could be easier if fixed terms are used but the review would fail to include studies that did not use such fixed terms). The authors may be able to explain and justify their approach.

(Authors' response) We thank the reviewer for their comment which is very similar to the 6th comment from Reviewer 1.

We have copied our response to their comment here:

'...we are interested in a broad range of health behaviours in combination, including conventional (e.g. smoking and alcohol) and more recently appreciated health behaviours (e.g. sleep). As we are not focussing on individual or single health behaviours for this review, we have not included search terms for single health behaviours as we believe that any article that investigates combinations of health behaviours would likely include one of the combined/combo terms from our search strategy in the title or abstract.'

Reviewer: 3

Comments to the Author

1. You are looking for studies that examine “lifestyle factor combinations” that include at least 3 lifestyle factors. It wasn’t clear to me whether the combinations had to be represented as a single “score”, or whether you would include a study that looked at the simultaneous effect of several lifestyle factors in a multivariable regression model.

(Authors’ response) Thank you for this comment and we apologise that this was not clear. We will include studies that include 3 or more lifestyle factors as a main exposure. We have not stipulated that this must be summarised as a single score.

2. This sentence is not clear. Stronger than what?: “For example, in a Scottish cohort, lower SES (measured by education level, social class, household income, and area-based deprivation) had stronger associations with alcohol-related hospital admissions and alcohol related-deaths at the same level of alcohol intake even after controlling for drinking patterns, smoking and BMI.”

(Authors’ response) We thank the reviewer for pointing this out. We have amended the sentence to read (line 57):

“For example, in a Scottish cohort, lower (as opposed to higher) SES (measured by education level, social class, household income, and area-based deprivation) had stronger associations with alcohol-related hospital admissions and alcohol related-deaths at the same level of alcohol intake even after controlling for drinking patterns, smoking and BMI.”

3. In Table 1, it seems that you saying that to be included, a study must not restrict participants based on age, sex, etc. This seems to contradict what you say later: “Therefore, because it was anticipated that few studies would fit the inclusion criteria, the population type was not restricted in order to identify as many studies as possible.”

(Authors’ response) We apologise that our language was not clear or consistent enough here. We mean that because there are likely to be few studies overall and in order to capture as much evidence as possible, we wanted to have the least restrictive population inclusion criteria. We are not saying that studies must not restrict in order to be included, we are saying that we will not exclude studies on the basis of cohorts’ population characteristics. We will include any study of a general population, regardless of study participants’ age, sex, or other sociodemographic characteristics.

To make this clearer, we have changed Table 1 ‘Population’:

“Studies of any general population type will be included. Eligibility will not be restricted by age, sex, or other sociodemographic characteristics. Cohort studies focussing on participants with an index condition/disease will be excluded.”

4. In table 2, I don’t know what #1, #2, etc mean. Is #10 is the final search you will use?

(Authors’ response) Yes, #10 will be the final search. We have added ‘Search’ to Table 2 in the column heading to make it clear that each number is a search and added (final search) to #10.

5. A potential limitation that should be mentioned is that different lifestyle factor combinations may represent totally different things, which will make it hard to draw conclusions. There is no reason to expect that the presence or absence (or direction) of an interaction between SES and a lifestyle factor combination would be the same for different lifestyle factors or lifestyle factor combinations, especially very different ones like smoking versus social participation.

(Authors’ response) We thank the reviewer for their comment. In response to reviewer 1’s 4th comment (please see above) regarding SES effect modification we have highlighted some of the potential mechanisms that might underlie any differential vulnerability where lower SES groups may be disproportionately vulnerable to the adverse effects of combinations of unhealthy lifestyle factors. We agree that different combinations of unhealthy lifestyle factors are likely to have different associations with adverse health outcomes and therefore, any influence of SES is likely to vary by

lifestyle combination as well. In fact, this is one of the potential areas of evidence that this review may highlight and we have actually added this as a strength rather than a limitation in the ‘Strengths and limitations of this study’ section under the Abstract:

“•Synthesising a broad evidence base to provide an overview of the potential influence of SES on associations between combinations of unhealthy lifestyle factors and adverse health outcomes could indicate which combinations of unhealthy lifestyle factors are associated with the greatest risks for lower SES groups.”

Related to this, we have added the following to the introduction as part of our response to reviewer 1’s 5th comment:

(line 35) “Furthermore, the additional risks associated with combinations of unhealthy lifestyle factors would motivate work to unpack which combinations have the highest risk. For example, if a combination of high sedentary time together with short sleep duration and poor diet is highlighted as particularly high risk then interventions could be targeted at this specific behavioural combination.”

And we have added the following to the Discussion (line 351):

“In addition, included studies may identify specific combinations of unhealthy lifestyle factors that pose the highest risks for lower SES groups. However, we suspect there is likely to be a lack of studies which identify the combinations that pose the greatest risk for lower SES groups and this may be one of the evidence gaps that this review identifies.”

VERSION 2 – REVIEW

REVIEWER	Eaton, Anne Memorial Sloan-Kettering Cancer Center, Biostatistics
REVIEW RETURNED	17-Dec-2020

GENERAL COMMENTS	When reading the original version, I thought you were potentially going to do a meta analysis that included different exposures in the same analysis, which was the basis for my comment 5. But now I see that you will do the meta analysis only across studies with the same exposure and outcome, and that you (realistically) state that meta analysis may not be possible. My other comments have been addressed.
--